# Long live the wasp: adult longevity in captive colonies of the eusocial paper wasp *Polistes canadensis* (L.)

Robin J. Southon[1], Emily F. Bell[1,2], Peter Graystock[1] and Seirian Sumner[1]

[1] School of Biological Sciences, University of Bristol, Bristol, UK
[2] Institute of Zoology, Zoological Society of London, London, UK

## ABSTRACT

Insects have been used as an exemplary model in studying longevity, from extrinsic mortality pressures to intrinsic senescence. In the highly eusocial insects, great degrees of variation in lifespan exist between morphological castes in relation to extreme divisions of labour, but of particular interest are the primitively eusocial insects. These species represent the ancestral beginnings of eusociality, in which castes are flexible and based on behaviour rather than morphology. Here we present data on the longevity of the primitively eusocial Neotropical paper wasp *P. canadensis*, in a captive setting removed of environmental hazards. Captive *Polistes canadensis* had an average lifespan of 193 ± 10.5 days; although this average is shorter than most bee and ant queens, one individual lived for 506 days in the lab—longer than most recorded wasps and bees. Natal colony variation in longevity does exist between *P. canadensis* colonies, possibly due to nutritional and genetic factors. This study provides a foundation for future investigations on the effects of intrinsic and extrinsic factors on longevity in primitively eusocial insects, as well as the relationship with natal group and cohort size.

Corresponding author
Peter Graystock,
peter@graystock.info

## INTRODUCTION

Death comes to all, yet many seemingly ordinary insects have evolved some of the most dramatic and extraordinary lifespans, delaying the call of death for remarkable periods (*Finch, 1990*). Variation in insect longevity spans from *Ephemera simulans* males that live as adults for just 1.6 days (*Carey, 2002*) to the ants *Pogonomyrmex owyheei* and *Lasius niger* whose queens can live up to 30 years (*Porter & Jorgensen, 1988*; *Hölldobler & Wilson, 1990*). Interestingly, eusocial insects such as ants, wasps, and bees feature heavily as examples of long-lived insects, but great variation exists not only between these species but also within species and even among genotypes. We understand little about the roles of ecology, evolution, life-history, and environment in generating variation in longevity in social insects, largely due to the difficulty of disentangling intrinsic life-span (hence-forth referred to as longevity) from survival (the abiotic and biotic environment pressures i.e., extrinsic mortality) on individuals (*Hölldobler & Wilson, 1990*; *Keller, 1998*; *Keeler, 1993*; *Giraldo & Traniello, 2014*).

Eusocial insects are one of the most dominant, prolific, and diverse groups of organisms on the planet (*Wilson, 1975*). Much of this groups' success is attributed to the division of labour within the colony in the form of castes, with few or a single reproductive individual (*queen)*, supported by tens to millions of non-reproductive individuals (*workers)* that forage, provision, and care for sibling brood (*Crespi, 1994*). Caste fate is primarily determined by environmental conditions, e.g., nutrition during larval development, and occasionally genetic biasing (*Oster & Wilson, 1978*; *Hölldobler & Wilson, 1990*; *Hughes et al., 2003*). Within species, variation in longevity can be pronounced between castes, with queens living as much as 100-fold longer than their related workers (e.g., general: *Kramer & Schaible, 2013*; *Lucas & Keller, 2014*; ants: *Hölldobler & Wilson, 1990*; *Keller & Genoud, 1997*; wasps: *Ridley, 1993*). This is a remarkable example of a how a single genome can display plasticity in aging (*Keller, 1998*; *Fjerdingstad & Crozier, 2006*; *Keeler, 1993*). Few individuals are selected to specialise in egg production and therefore colony survival is likely to be highly associated with and dependent on queen longevity (in the absence of reproductive succession, see *Bourke (2007)*). As a result, specialised egg layers are frequently protected from extrinsic pressures such as predation, for example the long-lived queens of the Harvester ant (*Pogonomyrmex owyheei*) live deep within the nest where they are sheltered (*Porter & Jorgensen, 1981*). Assuming there are costs associated with longevity (e.g., nutritional demands during larval development, development time), evolutionary theory would predict there would be selection for short lifespan in workers, and long-life span in queens, especially in highly eusocial species where colonies are large enough to support highly specialised, short-lived workers (*Evans, 1958*; *Carey, 2001*; *De Loof, 2011*; *Ferguson-Gow et al., 2014*). This has been shown to be the case with weaver ants in a protected lab environment whereby major workers (who take on more risky tasks) have a shorter intrinsic lifespan than minor workers who adopt less risky tasks (*Chapuisat & Keller, 2002*). The level of social complexity appears to be an important predictor of longevity in the eusocial insects. Castes are unlikely to have been selected for such differential longevity in the primitively eusocial species, where colonies are small, each worker is valuable, and survival of workers may be highly variable depending on the type or frequency of task each individual performs (*Strassmann, 1985*).

Between species, individual longevity is often correlated with mature colony size, as shown in several wasp (*Vespa spp.*) and ant (*Myrmica*, *Leptothorax*, *Solenopsis*, *Cataglyphis*) species (*Matsuura & Yamane, 1990*; *Schmid-Hempel, 1998*). In ants, at the colony level, the first worker brood are often physically smaller with shorter lifespans than those produced later in the colony cycle, such as nanitic workers of ants (*Porter & Tschinkel, 1986*). It is hypothesised that this may be due to the increase in levels of nutrition available to brood as the colony grows (*Oster & Wilson, 1978*; *Porter & Tschinkel, 1986*). As the colony grows, the ratio of workers to larvae often increases, the larvae will then benefit from increased quality and quantity of food, which can result in longer adult life-spans (e.g., in honey bees *Apis mellifera*: *de Groot, 1953*; *Eischen, 1982*). Conversely in the primitively eusocial paper wasp *Polistes exclamans*, *Strassmann (1985)* identified that late emerging workers survived less

time than early emerging workers in 1977 and 1978. This pattern was not detected in 1979 and its cause was suggested to be due to extrinsic factors.

Extrinsic factors such as parasitism, prey availability, and abiotic conditions can be powerful determinants of survival to wild individuals (*Gibo & Metcalf, 1978*; *Strassman, 1979*; *Strassmann, 1981*; *Tibbetts & Reeve, 2003*). To date there have been no studies on how individual longevity varies with colony size in primitively eusocial insects in the absence of such extrinsic factors. Based on the larval nutrition quality to adult longevity theory, we predict the same patterns will occur as in the highly eusocial species, since individuals emerging early in the colony cycle are subject to low worker:larvae ratio and therefore low quality nutrition. Conversely, those emerging late in the colony cycle experience high worker:larvae ratio and thus high quality nutrition (*Sumner et al., 2007*). Additionally, there may be a genetic link to longevity whereby some colonies are more likely to produce long living individuals than others, potentially due to heritable differences in feeding/hunting propensity (*VanRaden & Klaaskate, 1993*; *Herskind et al., 1996*; *Vollema & Groen, 1996*; *Klebanov et al., 2001*; *Sebastiani et al., 2012*; *Gems & Partridge, 2013*). If colony effects are important, we predict that variation in longevity will be greater between colonies than within colony, even in the face of group size variation. Finally, positive correlations between colony size and longevity may be due to social-behavioural and metabolic factors such as increased per capita work rate in small colonies (*Karsai & Wenzel, 1998*).

Here we provide primary data on longevity of females in captive colonies of the predatory and primitively eusocial Neotropical *P. canadensis* paper wasp. Primitively eusocial species, such as those of the paper wasp genus *Polistes*, have been used to extensively study the evolution of eusociality, with their lack of morphological differences and plasticity in caste (*Turillazzi & West-Eberhard, 1996*; although see *Hunt, 2006*). Although some studies have addressed the survivorship and colony phenology of some *Polistes* species (e.g., *O'Donnell & Jeanne, 1992b*; *Giannotti, 1997a*; *Clapperton & Dymock, 1997*), there remain few systematic attempts to quantify longevity, and variation of, in this well-studied genus. Many tropical Polistinae such as *P. canadensis*, although influenced by wet/dry seasonality in food abundance (and resulting colony productivity), mate and have colonies of various life-stages throughout the year (*Pickering , 1980*; *Clutton-Brock, 1991*). These study systems offer an excellent system for testing the influence of ecology, evolution, and environment on longevity, in the absence of seasonal curtailment of longevity found in temperate species. Studying insect lifespans in captivity, in the absence of predation and parasitism, is a valuable approach that allows us to quantify longevity in the absence of extrinsic mortality pressures (*Chapuisat & Keller, 2002*). We assess how natal colony size correlates with longevity under laboratory conditions and follow this up by comparing longevity in experimentally manipulated group sizes. Understanding variation in longevity in these organisms provides an excellent foundation to explore similar questions in the higher-order social vertebrates (*Carey, 2001*).

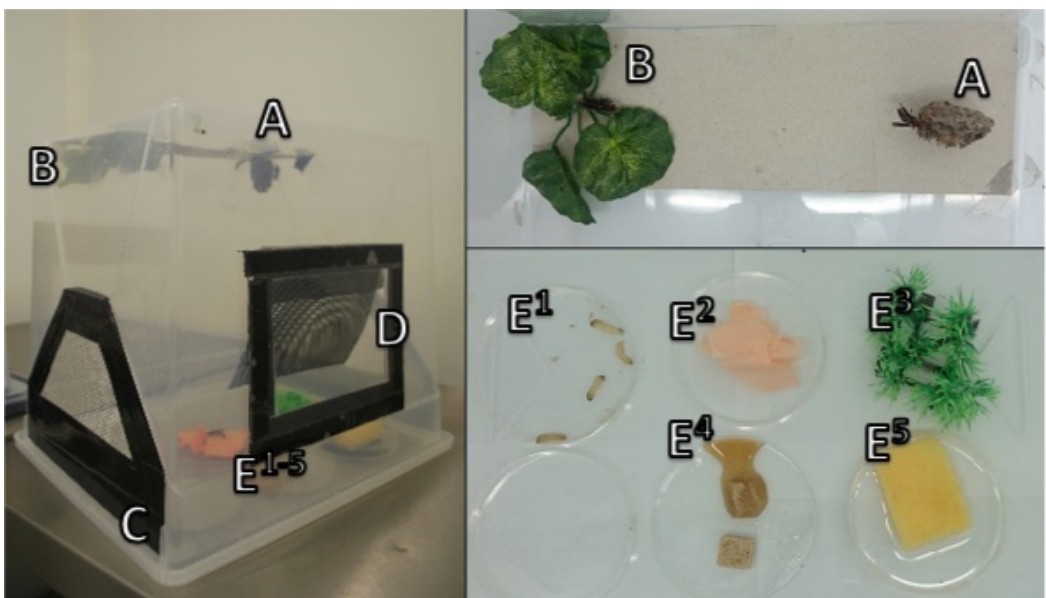

**Figure 1** **Captive housing of *P. canadensis* colonies.** A, Nest carton supported on reinforced celling with shade; B, artificial planting; C, ventilation; D, access hatch; E[1], food provisions; E[2], construction paper; E[3], artificial planting; E[4], liquid sugar cane; E[5], distilled water. Photographs by Robin Southon.

## METHODS

*Collection*: We collected ten colonies (M1–M10) of the paper wasp, *P. canadensis* from the Province of Colón in Republic of Panamá in August 2013 for transportation to the UK (9°24′03″N 79°52′11″W). Adult wasps were captured with full nest carton containing brood (eggs, larvae, and pupae) during dusk. The nest cartons and wasps were transferred to individual containers (15 cm x 15 cm x 15 cm) with wire mesh ventilation. Colonies were then provided with sugar solution and water *ad libitum* during transfer to the United Kingdom in luggage kept at ambient temperature. Turnaround from capture to settled maintenance in the laboratory was 48 h. To ensure that no colony was heavily infected with entomopathogens from the field, a subset (5 individuals per colony) of cadavers were placed in isolated petri dishes to observe any resulting sporulation of any infecting fungal entomopathogens. The common fungal agent *Aspergillus spp.* sporulated from 10% of these cadavers. *Aspergillius* is an opportunistic and largely ubiquitous fungus, commonly regarded as non-lethal to social insects unless under extreme stress or ingested at unnaturally high concentrations (*Bailey, 1968*; *Foley et al., 2014*).

*Maintenance*: Once in the UK, we housed nests in clear transparent acrylic containers 30 cm × 33 cm × 34 cm each with two 525 mm perimeter ventilation ducts (Fig. 1). The food provided consisted of liquid cane sugar and live wax moth larvae *Achroia grisella*, along with distilled water and nest-building materials (cardboard & paper) all were supplied *ad libitum*. All sugar and food was obtained in batches and haphazardly split between colonies to ensure equal food quality provided to the adults to prevent any longevity variability as a result of adult nutrition (*Johanowicz & Mitchell, 2000*; *Harvey et al., 2012*). In addition to food, in each nest-box we provided plastic artificial planting in the form of

a strip of 7 cm × 7 cm × 2.5 cm grass *sp.* and a 5 cm × 5cm × 5 cm plastic *Hedera sp.* for environmental enhancement to provide shelter from female aggression for males (*Polak, 2010*). The nest boxes were cleaned regularly with distilled water without disturbing wasps or nest. Natural conditions from the collection sites were mimicked with temperatures of $25 \pm 1\,°C$, $70 \pm 5\%$ relative humidity, and a light cycle of 12 h light (12 h dark).

*Data collection*: The colonies were surveyed three to four times a week and any deceased adults found immediately removed from the nest box. Total deaths per colony/cohort were tallied on a weekly basis and this recording method continued until all individuals were deceased. Whilst newly laid eggs were left in the nest for the adults to tend to, brood were eventually removed before pupation ensuring that only the original adult wasps captured from the wild were monitored for longevity and the colony/cohort sizes remained constant. This ensured all of the adult wasps developed under semi-natural conditions. Since all nests were collected from the same field site at the same time, local environmental conditions for development are controlled for as best as possible, though the colonies will differ from each other genetically.

As we do not know the eclosion date for each adult wasp, measures of longevity will be underestimates.

### Hypothesis 1: Adult longevity of female *P. canadensis* will show some positive correlation with the size of the natal group due to nutrition during colony development theory

Using the data generated from colonies M1–M7, we were able to quantify variance in longevity between colonies to determine whether colony identity explains variation in wasp longevity better than colony size. Using average survival per colony, correlations between initial colony size upon permanent laboratory setup were investigated.

### Hypothesis 2: Group size will correlate positively with mean female longevity in *P. canadensis* once the influence of colony genotype is controlled for

Three colonies (M8, M9, M10) were monitored for a period of 3 months, at which point their group sizes were of 28, 23 and 23 workers respectively. Each colony was then split, and randomly allocated between two new nest boxes lacking nest cartons, giving six new groups in total and consisting of 18, 13, 12, 8, 8 and 9 females. A non-natal male was also added to each the new colony nest boxes so that females had the opportunity to mate. All cohorts then started to build nest cartons and lay eggs suggesting mating may have occurred. The colonies were maintained as above with wasp deaths monitored weekly for 220 days, at which point all individuals were deceased.

*Statistical analyses:* Differences in adult survival were analysed using a Cox proportional hazards regression model where colony was used as a factor. Where differences in survival were found, we conducted pairwise comparisons between nests using Kaplan–Meier models with the Breslow $\chi^2$ statistic to highlight specific patterns between the colonies. Pearson product-moment correlations were carried out to look for patterns between

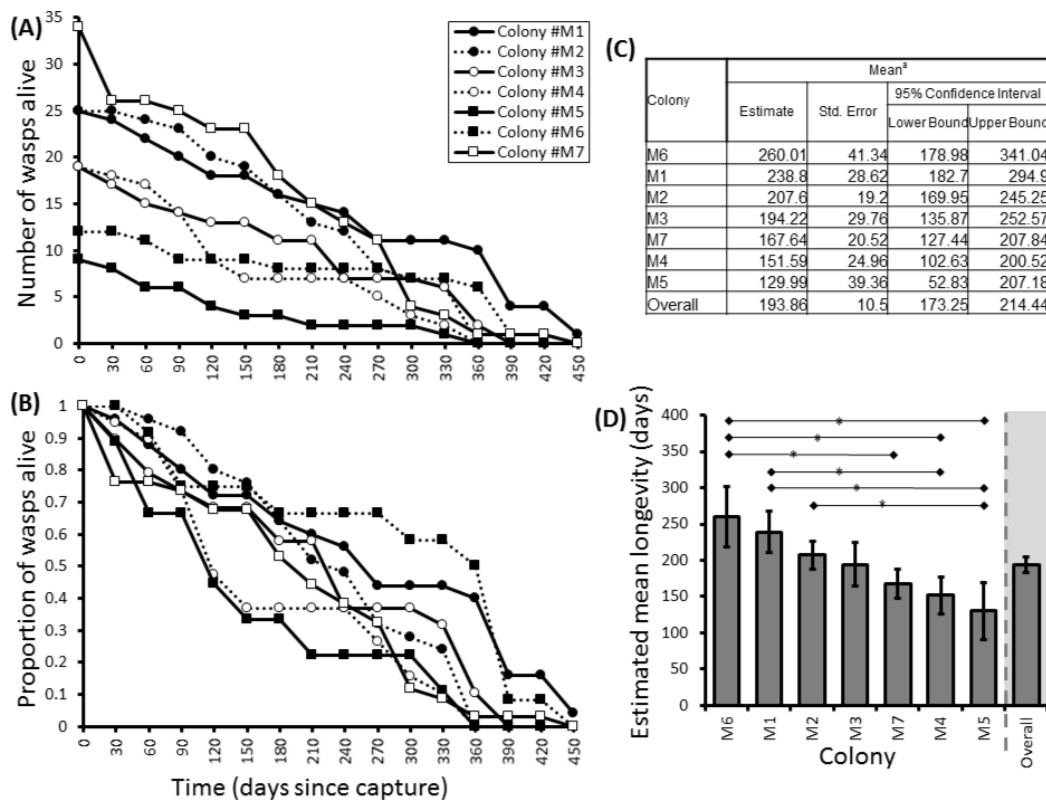

**Figure 2 Survival details of seven colonies of *P. canadensis* over 450 days.** Survival of adults in seven *Polistes canadensis* colonies shown as raw numbers (A) and proportions (B) over a period of 15 months post capture whilst maintained under laboratory conditions. Their longevity estimates of adult wasps for each colony as estimated by Kaplan–Meier survival analysis (C) with pairwise differences as calculated by the Breslow statistic shown by capped horizontal bars (D).

colony size and average colony longevity. All analyses were carried out in SPSS Statistics 21 (IBM, Armonk, NY, USA).

## RESULTS

### Hypothesis 1: Adult longevity of female *P. canadensis* will show some positive correlation with the size of the natal group

Here we found that 57% of adult *P. canadensis* colonies maintained in the lab can survive beyond 365 days with one individual living for 506 days, providing data on longevity for 143 wasps in total (Fig. 2). On average wasps lived for $193 \pm 10.5$ days with the oldest individual living for 506 days (Fig. 2 and Fig. S1). Colony identity has a significant influence on adult wasp longevity (Cox proportional hazard survival analyses Wald $= 17.134$, d.f. $= 6$, $P = 0.009$ (Fig. 2 and Table S2). There was no correlation between colony size and the colonies' average longevity ($r = 0.06$ $n = 7$ $P = 0.89$; Fig. 3A). Regular observations did not identify any behavioural differences between colonies. All colonies built nests, maintained social cohesion, and regular egg laying was observed throughout.

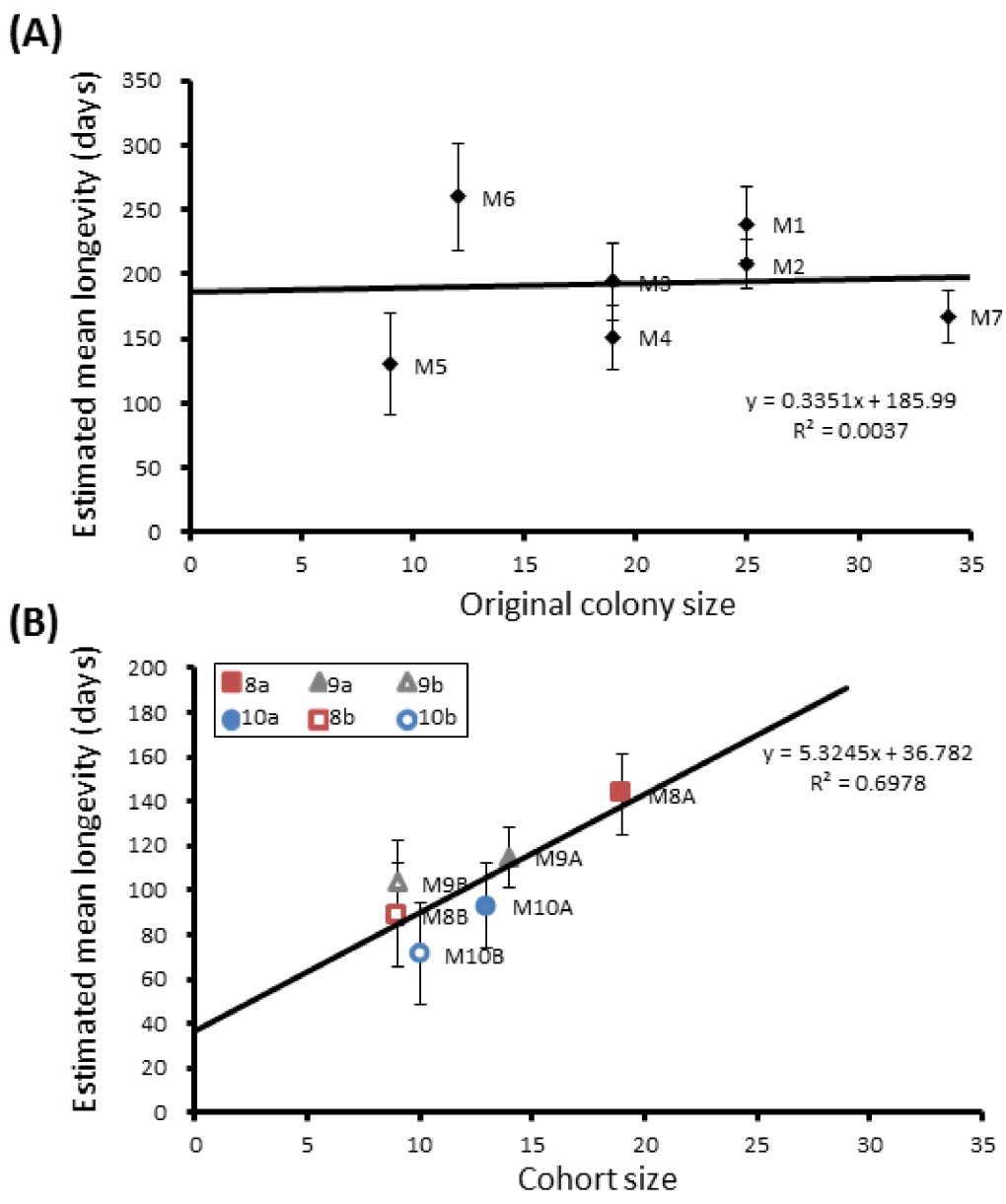

**Figure 3 Mean longevity per colony against colony size.** Correlations between colony size and mean longevity of adult *P. canadensis* when maintained in original colony (A) or when manipulated into cohorts of varying size (B). Standard error bars calculated by Kaplan–Meier model.

## Hypothesis 2: Group size will correlate positively with mean female longevity in *P. canadensis* once the influence of colony genotype is controlled for

Group sizes ranged from 9 (M5) to 34 (M7) wasps with average longevity within different nests ranging from $130 \pm 39.4$ days (in M5) up to $206 \pm 41.3$ days (in M6). Comparing the mean adult longevity of colonies M1–M7 against their original size gives no clear association (Fig. 3A). Colonies M8–M10 showed no difference in survival prior to splitting

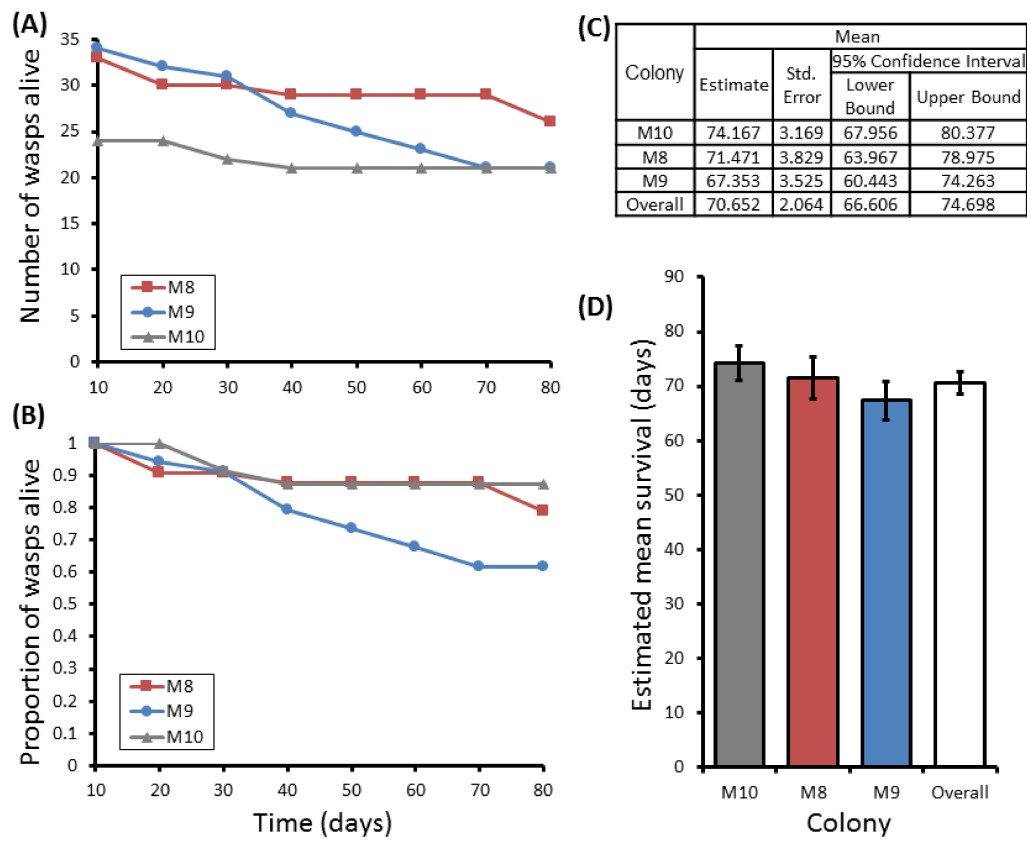

**Figure 4 Survival details of three colonies of *P. canadensis* over 80 days.** Survival of adults in three *P. canadensis* colonies (92 adults) shown as raw numbers (A) and proportions (B) over a period of 80 days post capture whilst maintained under laboratory conditions, along with the survival estimates for each colony as estimated by Kaplan–Meier survival analysis. (C) Estimations used in C-D are limited to the largest survival time due to censorship. Standard error bars in (D) calculated by Kaplan–Meier model.

(Wald $= 4.016$, d.f. $= 2$, $P = 0.134$; Fig. 4). However, after splitting into 6 cohorts of variable size, females exhibited significantly different longevities (Wald $= 12.544$, d.f. $= 5$, $p = 0.028$; Fig. 5). Cohorts from M9 (M9A & M9B) and from M10 (M10A & M10B) show no significant difference in adult longevity within natal colony identity ($\chi^2 = 0.173$, $P = 0.677$; $\chi^2 = 0.394$, $P = 0.530$ respectively; Table S3), cohorts from colony M8 (M8A & M8B) do differ from each other and are, incidentally, the 2 cohorts with the largest size difference ($\chi^2 = 3.829$, $P = 0.05$; Fig. 5). Group size shows a positive correlation with longevity ($r = 0.84$ $n = 6$ $P = 0.038$; Fig. 3B). Regular observations did not identify any behavioural differences between cohorts. All cohorts built nests, maintained social cohesion, and regular egg laying was observed throughout.

## DISCUSSION

Here we show that with an average lifespan of 193 days, *P. canadensis* have some of the longest lifespans of recorded wasps under laboratory conditions to date. The oldest wasp in our study lived for a staggering 506 days which is one of the longest living lab assisted, or wild recorded wasp—and most recorded wild and assisted bees, with the notable exception

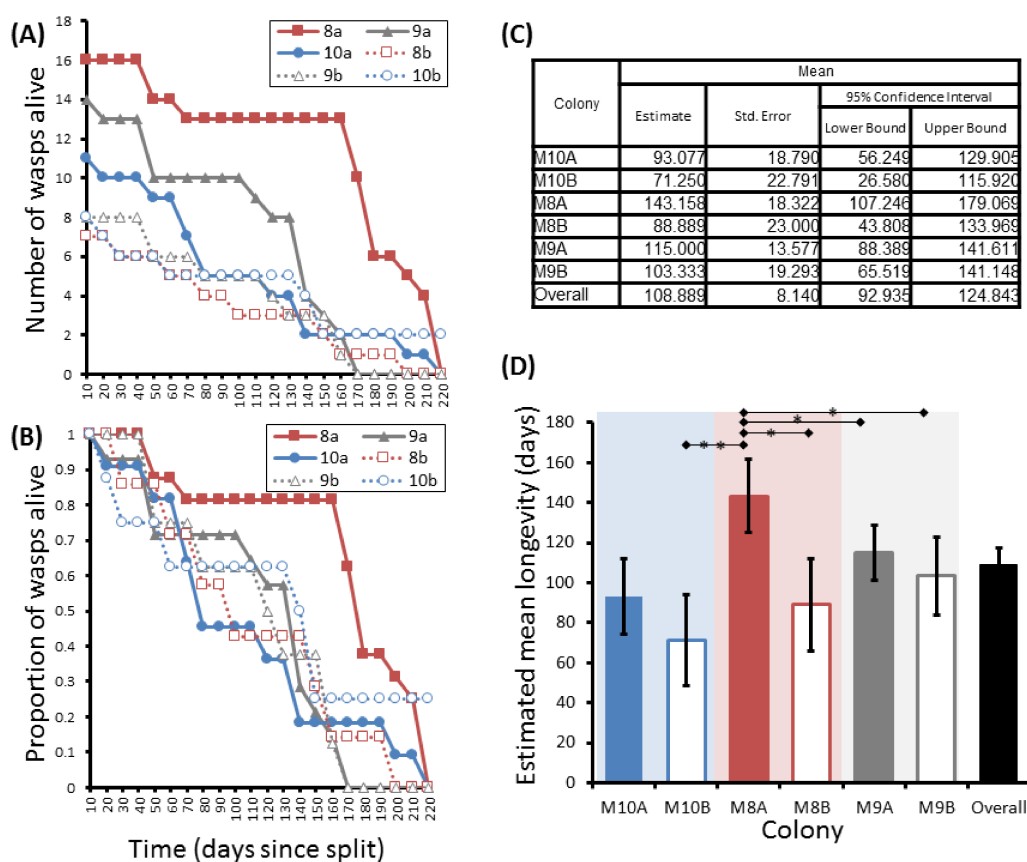

**Figure 5  Survival details of six cohorts of *P. canadensis* wasps from 3 natal colonies.** Survival of 72 adult *Polistes canadensis* in 6 conspecific groups split from 3 colonies, shown as raw numbers (A) and proportions (B) over a period of 220 days post split whilst maintained under laboratory conditions. The survival estimates of these as estimated by Kaplan–Meier survival analysis are shown (C) with pairwise differences as calculated by the Breslow statistic shown by capped horizontal bars (D).

of assisted honeybee queens tending to live around live around 3,000 days (Fig. 6 and Table S3). The natal group was a significant predictor of longevity with wasps living on average between 260 days and 130 days depending on nest, though the size of the natal groups had no correlation with longevity. When natal groups were split in to varying cohort sizes, the largest cohort lived the longest and a correlation between longevity and cohort size was identified. We discuss the implications of these data in the context of other species and ecology and evolution of eusociality.

We observed large cohorts of long lived individuals in multiple colonies, although we cannot say whether individuals were of queen, worker, or in a quiescence (*Hunt, 2006*) status/state. In the most general sense, eusocial structure is based on behaviour rather than the physiological constraints observed in more highly eusocial hymenopterans such as honey bees and many ants, which incidentally display the largest longevity disparities between castes. Though there are no directly comparable studies that use *Polistes* in a lab setting such as this study, the lifespan estimates on workers of wild or assisted *Polistes* species tends to be approximately one month, with the longest average life span

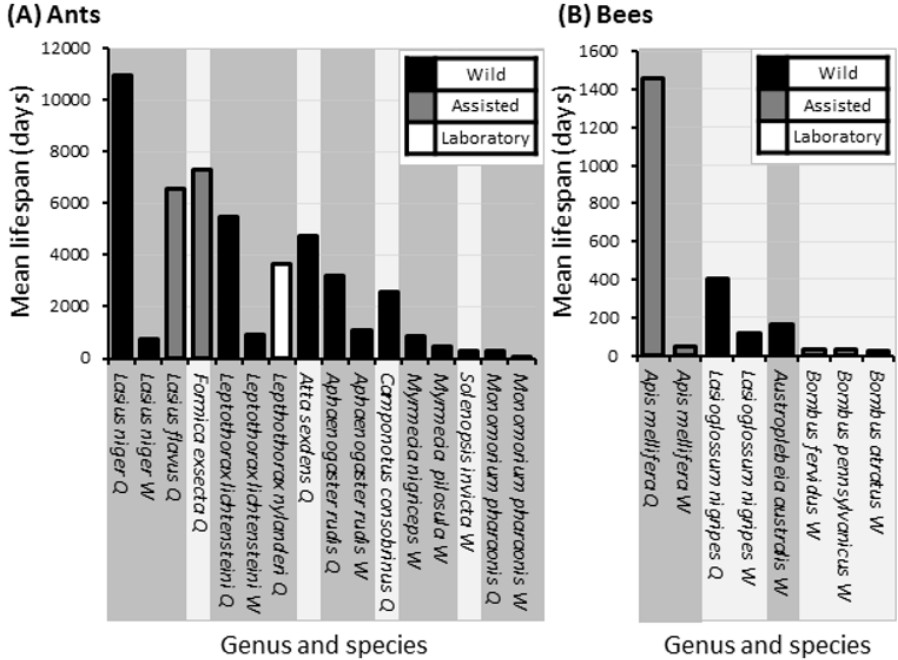

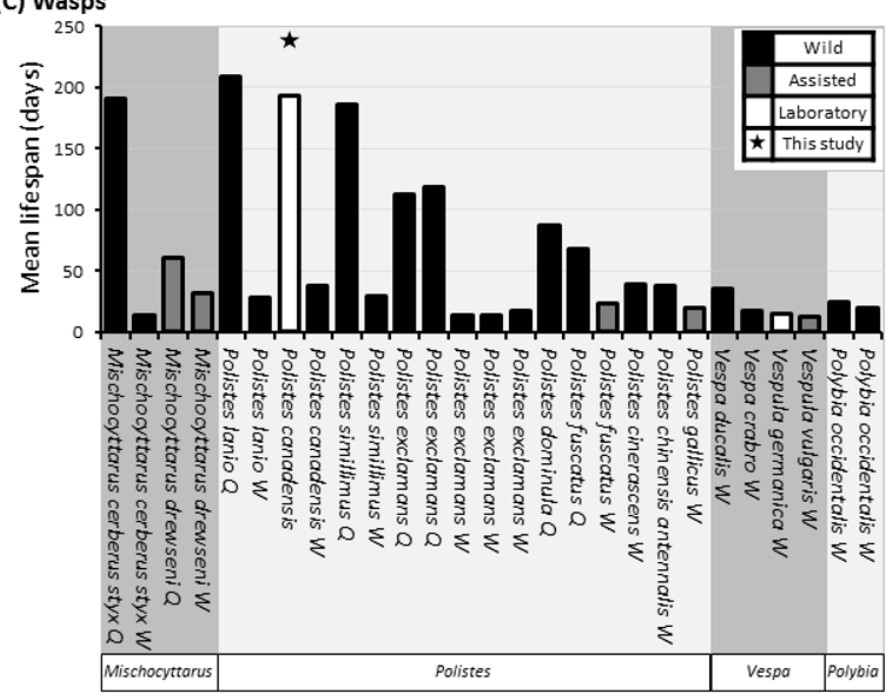

**Figure 6 Comparative mean lifespans in ants, bees and wasps.** Comparison of lifespans in a selections of ants (A), bees (B) and all records of wasps (C), highlighting whether data is attained from wild (Black bars) or assisted (grey bars) colonies with unknowns also included (grey bars). Queen lifespans in minimum age, as most studies/literature start from colony creation. Data compiled from the result of this study (above column) and those found following literature search (continued on next page...)

**Figure 6 (...continued)**
(*Pardi, 1948*; *Michener, 1969*; *West-Eberhard, 1969*; *Matsuura, 1971*; *Wilson, 1971*; *Spradbery, 1973*; *Miyano, 1980*; *Haskins & Haskins, 1980*; *Akre, 1982*; *Strassmann, 1985*; *Goldblatt & Fell, 1987*; *Dazhi & Yunzhen, 1989*; *Hölldobler & Wilson, 1990*; *Pamilo, 1991*; *O'Donnell & Jeanne, 1992a*; *Giannotti & Machado, 1994a*; *Giannotti & Machado, 1994b*; *Keller, 1998*; *da Silva-Matos & Garófalo, 2000*; *Page Jr & Peng, 2001*; *Gamboa, Greig & Thom, 2002*; *Jemielity et al., 2005*; *Hurd, Jeanne & Nordheim, 2007*; *Archer, 2012*; *Giannotti, 2012*; *Torres, Gianotti & Antonialli-Jr, 2013*; *Halcroft, Haigh & Spooner-Hart, 2013*; *Jeanne, 1975*).

being 37 days in *P. lanio* and shortest life span being 14 days in *P. exclamans* (*Miyano, 1980*; *Strassmann, 1985*; *Giannotti & Machado, 1994b*; *Giannotti, 1997a*; *Giannotti, 2012*; *Gamboa, Greig & Thom, 2002*; *Torres, Gianotti & Antonialli-Jr, 2013*).

The average lifespan of egg laying *Polistes* wasps can range from a maximum of 209 days in wild *Polistes lanio* queens down to 66 days in wild queens of *Polistes fuscatus* wasps (*Giannotti & Machado, 1994b*; *Gamboa, Greig & Thom, 2002*). Whilst there is an observation from Phil Rau (*West-Eberhard, 1969*) of a marked wild temperate *Polistes* queen living for approximately 2 years, it seems tropical species may have some of the longest lifespans, although further empirical studies are needed to tease apart seasonal affects. In each case, the studies were carried out on wild *Polistes* colonies and so cannot account for extrinsic mortality (*Strassmann, 1985*; *Giannotti & Machado, 1994b*). The life-span of wasps in more highly eusocial species ranges from 1,000 days in queens of some wild *Vespa spp.* to 14.5 days in lab maintained *Vespula germanica* and *Vespula consobrina* (*Akre, 1982*; *Dazhi & Yunzhen, 1989*; *Hölldobler & Wilson, 1990*).

On average there is a positive correlation between the maximum lifespan of eusocial queens and the degree of eusociality displayed by that species (*Carey, 2001*; *Kramer & Schaible, 2013*) and differences found among species tend to be due to extrinsic mortality (*Keller, 1998*). Here we cannot differentiate between egg layers and helpers in our primitively eusocial species, yet evolutionary theory dictates that when a colony is small and the lifespans of both the reproductive individuals and helpers are equal, the helpers will resist evolutionary specialisation to workers as that would ultimately reduce their direct fitness potential (*Alexander, Noonan & Crespi, 1991*). We may therefore expect to find similar lifespans in both egg layers and workers in *P. canadensis*. However differences in lifespan are observed between helpers and egg layers in other *Polistes* species (Fig. 6). These studies use wild species though and describe the extrinsic mortality, unlike our study which had minimal extrinsic pressures.

Colony identity was a clear predictor of wasp longevity. All colonies were collected at the same time from the same field site, in which adults on all nests would have shared the same developmental and environmental conditions. The potential causes for the effect of colony identity could be: (1) genetic differences between the colonies. Genetic influences on longevity have been found in a number of model species from mammals to nematodes and insects (*VanRaden & Klaaskate, 1993*; *Herskind et al., 1996*; *Vollema & Groen, 1996*; *Klebanov et al., 2001*; *Sebastiani et al., 2012*; *Gems & Partridge, 2013*) and evidence for heritability of increased longevity in the fruit fly and honey bee has been observed (*Rinderer, Collins & Brown, 1983*; *Luckinbill & Clare, 1985*) with some gene

expression patterns being associated with longevity in queen honey bees (*Corona et al., 2005*). (2) Queen 'quality' which can be the result of extrinsic or intrinsic factors. Variation in fecundity of reproductive and dominance over other individuals in a colony is known as queen quality and this can vary between queens (*Harris & Beggs, 1995*; *Liebig, Monnin & Turillazzi, 2005*; *Holman, 2012*). This queen quality variation can be inherited (*Rinderer & Sylvester, 1978*; *Corona et al., 2005*) or driven by environmental factors (*Hatch, Tarpy & Fletcher, 1999*; *Tarpy et al., 2011*). (3) Unobserved differences in extrinsic factors that the nests had experienced before collection. Since the colonies were not monitored for their entire life cycle, there is the possibility that something affected each one differently in order to cause varying longevity within their workers. What we can conclude is that although colony genotype was a predictor of longevity in the adult wasps, this did not correlate with wasp group size unless the size was manipulated. This suggests that the colony influences are greater than those of group size, and whilst an overall correlation between manipulated group size and longevity was identified, only one out of three split colonies displayed this trend. This suggests that the explanation that larger colonies produce longer lived workers due to enhanced nutrition during larval development is not a major component.

To investigate the underlying variation in longevity in eusocial insects, data from captive colonies of a range of eusocial insects is a useful tool and can help uncover variation in investment for longevity based on extrinsic factors faced by a given species, individual, or caste (*Chapuisat & Keller, 2002*). Longevity studies on predatory eusocial insects such as wasps are underrepresented in the literature, and while several excellent studies have been identified, no studies have followed maintained *Polistes* in a protected lab environment. Here, for the first time we quantify longevity of adult *P. canadensis* in the absence of extrinsic mortality and provide some support for the link between group size and adult longevity but show that natal origin (i.e., genotype) is a more powerful predictor. Our results suggest that predictions founded on previous research using higher eusocial species such as honey bees may not be relevant to primitively eusocial species since their caste/fate is not fixed during development. A particular challenge will be for future studies to also control for all of the described extrinsic and intrinsic factors such as wild nest site condition and the presence of symbionts.

## ACKNOWLEDGEMENTS

Thanks to Smithsonian Tropical Research Institute, particularly Jorge Morales and the staff at Galeta Point, and field-assistant Daniel Fabbro. Research was made possible following ANAM research permits SE/A-20-12 and SE/A-55-13. Thanks to Ian Warren & Jacob Podesta for wasp husbandry.

### Funding

Research was funded by a NERC studentship to EB & RS, and a STRI pre-doctoral research fellowship to EB. The funders had no role in study design, data collection and analysis, decision to publish, or preparation of the manuscript.

## Grant Disclosures

The following grant information was disclosed by the authors:

NERC.

STRI.

## Competing Interests

The authors declare there are no competing interests.

## Author Contributions

- Robin J. Southon and Emily F. Bell conceived and designed the experiments, performed the experiments, contributed reagents/materials/analysis tools, wrote the paper, reviewed drafts of the paper.
- Peter Graystock conceived and designed the experiments, analyzed the data, wrote the paper, prepared figures and/or tables, reviewed drafts of the paper.
- Seirian Sumner conceived and designed the experiments, contributed reagents/materials/analysis tools, wrote the paper, reviewed drafts of the paper.

## Field Study Permissions

The following information was supplied relating to field study approvals (i.e., approving body and any reference numbers):

ANAM research permits SE/A-20-12 and SE/A-55-13.

## Supplemental Information

Supplemental information for this article can be found online at http://dx.doi.org/10.7717/peerj.848#supplemental-information.

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
