# Peer review of "Long live the wasp: adult longevity in captive colonies of the eusocial paper wasp Polistes canadensis (L.)"

_PeerJ, doi:10.7717/peerj.848_

## Round 0.1 · original submission · Minor Revisions

· Academic Editor

Minor Revisions

Please revise according to the comments of the reviewers.

Reviewer 1 ·

Basic reporting

comments give in other section

Experimental design

comments given in other section

Validity of the findings

comments given in other section

Additional comments

General comments:

The study seeks to determine the longevity of Polistes canadensis wasps in captivity. Longevity is recorded from wasps captured on wild nests that were subsequently brought to the lab. The data are presented adequately, though the figure numbering should be revised. Additional information on the nests that were captured and what the wasps did in captivity – were they working on their nests and rearing brood? If so how did the level of work relate to subsequent longevity? Did the wasps that lived longer tend to remain on the nest and care for the colony or simply leave the nest and be quiescent in the boxes? In order to better understand what the results mean data that can answer such questions are needed.

Specific comments:
In the abstract “primitively eusocial orders”, the word orders should be changed as it has the meaning of order of insects (e.g. Hymenoptera), which is not really a primitively eusocial order. Perhaps use species or lineages instead of orders.

In the abstract, “in comparisons to other eusocial wasps’ “ There is no clear comparison made in the sentence and it is unclear why wasps is made possessive.

L20. Genotype should be plural

L44. “selected for such differential”…word missing?

L75. There have in fact been a number of published studies of the survivorship and colony phenology in Polistes from both tropical and temperate species. They differ form this study in that they were survivorship studies rather than focusing on intrinsic longevity, however, it would be more scholarly to give credit to those other studies here or at least some other place in the introduction.

L137. It may be of interest to the authors that a report from Phil Rau of a queen being found at the site of her nest from the previous year (i.e. a female born in August or September of year zero, nesting through year 1 and appearing in april of year 2) is recorded on page 26 of West Ebhard’s “The Social Biology of Polistine Wasps”

L145. Figure 7 should be called figure 4, presuming that PeerJ follows the convention of figures being numbered in the order of reference in the text. This occurs for a number of other figures too. Please correct.

L147. Colony identity, while related to genotype, is far from a good proxy for genotype. We have not been provided any other information about the colonies in terms of their relative stage of maturity, number of cells, were they queen-right, the brood/worker ratio, etc. It seems that any of these factors could well influence the subsequent colony longevity as much if not more than genotype. If such data is available, it would be useful for the authors to analyze it and present which factors, if any, are associated with longevity. If they are not available, they authors need to very much tone down the interpretation of the colony level effect as one that is based on genotype.

L 155. Give stats in the text.

·

Basic reporting

This paper is about how long wasps live in captivity, using a tropical species that has no morphological castes. Besides nice work on this species, the paper includes a figure and a summary of longevity of workers and queens in other social insects.

Overall the paper is well written. However, a major problem in the introduction is the number of generalizations based on one or two unnamed species. The writing needs to be much more precise. There are issues with how the comparative data are use, with what a hypothesis is, and with how the data are interpreted. These issues are detailed below and should be easy to address in a revision. The paper is a useful contribution to our understanding of wasp longevity.

Lacking from the discussion is any treatment of what others have said about wasp longevity, or social insect longevity in general. The paper needs additional discussion to put these findings in a conceptual framework.

Experimental design

The design seems adequate, but more detail would be good. When did the wasps have nests and brood? Always? What happened to the newly emerged workers? Were there dead wasps potentially lying in the bottom of the cage?

See line by line comments in last section.

Validity of the findings

The comparison to other species confuses lab and field studies. There is no mention of other studies and their conceptual findings in the discussion, past the summary analysis of other study's results.

Additional comments

All the comments are here, line by line. I hope they are useful to you in revising this interesting study to make it have greater impact.

Line 39 “deep within the nest” is not a concept for many social insects, including Polistes. Make sure to be clear as to which species your sentences refer to.

Line 45 missing a comma after “small.”

Line 50 The claim that first worker brood have a shorter lifespan is not true for all species. In Polistes exclamans, for example, the first emerging workers can live a very long time. Here and throughout, be careful about the generalizations. What species are the cited papers referring to?

Line 48 on You use “and” often as a connector. It is a weak one. Either make it two sentences, or use a connector that gives more indication of what kind of connection you are making.

Line 57 where you say “e.g. in honey bees” is good and is a construction you should use more to avoid inaccurate generalizations.

Line 57 You say there have been no studies of worker longevity and colony size, then make predictions about time of year and worker survival. The Strassmann 1985 paper addresses exactly that issue, finding longer lives early in two years.

Line 63 is a run on sentence. Start a new sentence with “For example…” This paragraph is not very clear. Explain the arguments more specifically.

Line 73 caste in Polistes is not so simple. This reference is nearly 20 years old. I basically agree with the general statement, but there are differences that may be due to conditions on emergence, for example. Plasticity is maintained in some but not other species. The statement is too flippant.

Line 75 on tropical species and seasonality. Wow, this is a huge generalization. What was Clutton-Brock referring to? Many tropical species have a very strong seasonal pattern around wet and dry seasons. Many social wasps essentially shut down in the dry season. But not all do. Some tropical rainforests are nearly aseasonal. But this statement is surely incorrect.

Line 79. I agree that it is interesting to look at longevity in captivity, but it is less of a null than you might think. There are certainly fungi and bacteria in the cages and these might impact longevity. How much exercise and all kinds of other things can also impact cage longevity, so one cage study is not strictly comparable to others. Comparing longevity among colonies and for different group sizes is fine because these are controlled.

Lines 80 to 85 refer to hypotheses, but no hypotheses are given. A hypothesis generally takes the following form: A increases (or decreases, or happens) because of B for biological reason C. There is nothing like that here. Either reword to take hypothesis out of it, or put these statements in the form of actual hypotheses.

Line 87 forward. I prefer active voice. This is dismal to read, all passive.

Line 88 Panama is a big place. Give exact location of collection with longitude and latitude.

Line 91 transfer to UK was at ambient temperature? Checked luggage or carry on? This is interesting.

Line 90 and 94, seems like cm would be better for the cage measures than mm.

Line 97 do you mean randomly which is something specific using a random number generator, or haphazardly which is less precise?

Line 99 What is artificial planting? You put a plastic plant in there? What did it look like?

Line 101 did you clean with just water or something else?

Line 104 How often did you check the nests? This makes it sound like once a week, which would mean a dead wasp could lie in the cage rotting for nearly a week. Otherwise, why would you not record deaths daily?

Line 101 Your nests had pupae, so as these hatched you could have known their ages. Also, pupae could have added to the number there, so it is important to keep track of them.

Line 112 to 122. The first of these is close to a hypothesis. You should say correlates positively or negatively. The next two are not hypotheses because they do not give a direction for the change or a reason, only say there is a relationship between two things.

Line 127 Is one male sufficient to give a mating opportunity to 8 to 18 females? It is not clear in this treatment if the females had any nest carton or not.

Line 130 Tell what statistical package you used.

Line 137 I would first tell what you found for your wasps, then talk about the literature, paying particular attention to lab studies, which are more comparable. You don’t really do any analyses here. You don’t say in the figure what the white bars are. I’m guessing they are the lab studies? I like the comparison, but in the absence of any analysis, I would say this part really belongs in the discussion. Could you also glean any reasons for the patterns from the other studies? In short, you don’t really answer this hypothesis with this study. If you moved this comparison to the discussion, you could talk more about what other people thought about their studies, in particular all the work of Laurent Keller.

Line 139 awkward wording “so unable” lacks a verb.

Line 149 it is always true that colony will have an impact on things, whatever the other variable is. This isn’t a hypothesis and this isn’t a test. Generally what one does is show colony has an effect, then remove that effect from other studies. So this could go first as a result and then be controlled for in other things. Above you give an average longevity, but don’t make it clear how you did the average, whether you took an average for each colony, then a global average or not, for example.

Lines 152 to 165 is unclear. Reformulate the hypothesis to have a directional prediction and a reason, then think carefully about the order of what to tell us. Does the last sentence mean the “positive trend” is not significant?

Line 166 to 171. How many other reports of longevity are you comparing to? You jump from comparing among species to within in the same paragraph. Best to do one then the other.

Line 175 You are comparing your lab results to other results that are taken from the field. In every place, you need to separate these out.

Line 178 When you start comparing to queens, it warrants a new paragraph from the one talking about workers. Again, field to lab comparisons are not valid.

Line 207 on. You can get some things from lab studies, but be careful before you assume that this is the true innate longevity since the lab environment is also biotic. This last paragraph doesn’t really say very much and is not specific enough to your results. It also overgeneralizes.

Figure 1. The photo is quite unclear. For example, I still can’t tell what the “artificial planting” is. C, D, and E are all on the tape. A drawn cartoon might be more clear.

Figure 2. This figure would be better organized to have workers and queens from the same species together, and a standard deviation if available. It should be accompanied by a supplemental table giving lifespan, a measure of variation like standard deviation, sample size, conditions, and reference.

---

## Round 0.2 · accepted · Accept

· Academic Editor

Accept

Thank you for your conscientious revisions.